# A family of oxychloride amorphous solid electrolytes for long-cycling all-solid-state lithium batteries

Shumin Zhang[1,2,6], Feipeng Zhao [1,6], Jiatang Chen [2], Jiamin Fu[1,2], Jing Luo[1], Sandamini H. Alahakoon[2], Lo-Yueh Chang[3], Renfei Feng [4], Mohsen Shakouri [4], Jianwen Liang[1], Yang Zhao [1], Xiaona Li[1], Le He [5], Yining Huang [2], Tsun-Kong Sham [2] ✉ & Xueliang Sun [1] ✉

Solid electrolyte is vital to ensure all-solid-state batteries with improved safety, long cyclability, and feasibility at different temperatures. Herein, we report a new family of amorphous solid electrolytes, $xLi_2O \cdot MCl_y$ (M = Ta or Hf, $0.8 \leq x \leq 2$, y = 5 or 4). $xLi_2O \cdot MCl_y$ amorphous solid electrolytes can achieve desirable ionic conductivities up to $6.6 \times 10^{-3}$ S cm$^{-1}$ at 25 °C, which is one of the highest values among all the reported amorphous solid electrolytes and comparable to those of the popular crystalline ones. The mixed-anion structural models of $xLi_2O \cdot MCl_y$ amorphous SEs are well established and correlated to the ionic conductivities. It is found that the oxygen-jointed anion networks with abundant terminal chlorines in $xLi_2O \cdot MCl_y$ amorphous solid electrolytes play an important role for the fast Li-ion conduction. More importantly, all-solid-state batteries using the amorphous solid electrolytes show excellent electrochemical performance at both 25 °C and −10 °C. Long cycle life (more than 2400 times of charging and discharging) can be achieved for all-solid-state batteries using the $xLi_2O \cdot TaCl_5$ amorphous solid electrolyte at 400 mA g$^{-1}$, demonstrating vast application prospects of the oxychloride amorphous solid electrolytes.

Along with the fast growing market of rechargeable electric vehicles (REVs), the development of all-solid-state batteries (ASSBs) is of high expectation due to their promises of safety, reliability, and high energy density[1,2]. A key component for ASSBs is solid electrolyte (SE) which can potentially enable the use of high-voltage cathodes and Li metal anode to boost the energy density[3,4]. One of the essential requirements for a favorable SE is high ionic conductivity. Crystalline SEs with long-range ordered structures have shown continuous and fast Li-ion conduction. For example, representative sulfide-based SEs, such as Li

Argyodites[5,6] and $Li_{10}GeP_2S_{12}$ (LGPS)-type[7,8], exhibit attractive ionic conductivities in the order of $10^{-2}$ S cm$^{-1}$. Other types of crystalline SEs including oxide-based SEs (e.g., perovskite-type[9], sodium superionic conductor (NASICON)-type[10], and garnet-type[11,12]) and halide-based SEs (e.g., Li-M-Cl system, M = Y, In, Sc[13–17]) also demonstrate good conductivities of $10^{-4}$–$10^{-3}$ S cm$^{-1}$. While the ion conduction mechanisms of crystalline SEs have been widely studied to provide guidance for the search of new superionic conductors, some amorphous SEs also show good potentials but are less studied[18].

[1]Department of Mechanical and Materials Engineering, University of Western Ontario, London, ON N6A 5B9, Canada. [2]Department of Chemistry, University of Western Ontario, London, ON N6A 5B7, Canada. [3]National Synchrotron Radiation Research Centre, 101 Hsin-Ann Road, Hsinchu 30076, Taiwan. [4]Canadian Light Source Inc., University of Saskatchewan, Saskatoon, Saskatchewan S7N 2V3, Canada. [5]Institute of Functional Nano & Soft Materials (FUNSOM), Jiangsu Key Laboratory for Carbon-Based Functional Materials & Devices, Soochow University, Suzhou, PR China. [6]These authors contributed equally: Shumin Zhang, Feipeng Zhao. ✉e-mail: tsham@uwo.ca; xsun9@uwo.ca

Amorphous SEs present the primary advantages of softness, easy fabrication, low grain boundaries, wider compositional variations, and isotropic ionic conduction[19], which are expected to compensate the drawbacks of some crystalline SEs with high grain boundary resistance, poor processibility, and high cost. Despite the diligent efforts, the research for amorphous SEs has been proceeding slowly. One major challenge is that the ionic conductivities of amorphous SEs are generally lower than those of the typical crystalline SEs. Another challenge is lacking long-range periodicity that makes it difficult to understand the ion conduction mechanism in amorphous materials. There are very limited established universal theories for structure modeling and ionic diffusivity prediction for amorphous materials[20]. Different types of reported amorphous SEs have their own advantages and disadvantages. Early researches for amorphous SEs have been reported since the 1960s[19]. The sulfide-based amorphous SEs, such as $Li_2S$-$P_2S_5$[21] and $Li_2S$-$SiS_2$[22], show decent ionic conductivities around $10^{-4}$ S cm$^{-1}$, but their narrow electrochemical stability window (1.5–2.5 V vs. Li$^+$/Li)[23,24] and poor electrode compatibility[25] significantly limit their application in ASSBs. Oxide-based amorphous SEs (such as $Li_2O$-$MO_x$, M = Si, B, P, Ge, etc.[26-29], and lithium phosphorus oxynitride[30]) exhibit improved (electro)chemical stability compared to the sulfide compounds, but their poor ionic conductivities of $10^{-9}$–$10^{-6}$ S cm$^{-1}$ at room temperature (RT, 25 °C) are far away from the benchmark ($10^{-3}$ S/cm) for bulk-type ASSBs[31]. Nevertheless, if amorphous SEs were to be competitive with the crystalline SEs, both high conductivity and good compatibility with favorable layered oxide cathodes are required.

Herein, we report a family of lithium-based oxychloride amorphous SEs ($xLi_2O$-$MCl_y$, M = Ta or Hf, $0.8 \leq x \leq 2$, y = 5 or 4). The newly developed SEs display several desirable features compared to the existing SEs. (1) Facile synthesis method. One-step ball-milling method can easily yield the desired products in amorphous state. (2) High ionic conductivity. The optimized $1.6Li_2O$-$TaCl_5$ amorphous SE possesses a high ionic conductivity of $6.6 \times 10^{-3}$ S cm$^{-1}$ at 25 °C, surpassing most of the other amorphous SEs. Similarly high ionic conductivities can be maintained within an amorphous formation region (x = 1.1–1.8) for $xLi_2O$-$TaCl_5$. X-ray absorption spectroscopy (XAS) and other advanced techniques are combined to clarify that Ta-centered trigonal bipyramids with rich terminal chlorines are predominant for fast Li-ion diffusion. The optimized $1.5Li_2O$-$HfCl_4$ amorphous SE also exhibits a good ionic conductivity of $1.97 \times 10^{-3}$ S cm$^{-1}$ at 25 °C. (3) Outstanding electrochemical performance. The $xLi_2O$-$MCl_y$ amorphous SEs have good compatibility with different favorable oxide cathodes without any additional cathode coatings. ASSBs using $xLi_2O$-$MCl_y$ amorphous SEs showed promising long-life cycling performance at both 25 °C and −10 °C.

## Results

### Preparation of the $xLi_2O$-$MCl_y$ materials

The $xLi_2O$-$TaCl_5$ amorphous SEs were prepared by simply ball-milling $Li_2O$ and $TaCl_5$ at various stoichiometric ratios. Lab-based X-ray diffraction (XRD) patterns of the as-prepared $xLi_2O$-$TaCl_5$ ($1 \leq x \leq 2$) are shown in Fig. 1a. When x = 1, the pattern was generally amorphous with a few unknown impurities. Slightly increasing the molar ratio of $Li_2O$/$TaCl_5$ to 1.1 led to a completely amorphous feature (see zoom-in figures in Supplementary Fig. 1). Interestingly, further increase of the $Li_2O$ content ($x \leq 1.8$) did not change the amorphous nature of $xLi_2O$-$TaCl_5$ SEs. As proved by synchrotron-based 2D diffraction patterns in Fig. 1b, the similar vague halos were recorded for the selected $1.1Li_2O$-$TaCl_5$, $1.6Li_2O$-$TaCl_5$, and $1.8Li_2O$-$TaCl_5$ samples, indicating amorphous states for the $xLi_2O$-$TaCl_5$ SEs. Therefore, the amorphous formation region for the $xLi_2O$-$TaCl_5$ system was identified as $1.1 \leq x \leq 1.8$. For a higher feeding ratio of $Li_2O$ ($x \geq 1.9$), LiCl impurity appeared. SEM showed the continuous and compact surface morphology of one compound ($1.6Li_2O$-$TaCl_5$ pellet) in Supplementary Fig. 2, which was sharply contrast to other crystalline SE under the similar ball-milling conditions. In addition to the $xLi_2O$-$TaCl_5$, lithium-based oxychloride amorphous SEs can be extended to the other systems involving high-valence transition metal chlorides, for example, using Hf$^{4+}$ instead of Ta$^{5+}$. Figure 1c depicts the lab-based XRD patterns for a series of $xLi_2O$-$HfCl_4$ ($0.8 \leq x \leq 2$) samples. In comparison with the $xLi_2O$-$TaCl_5$ amorphous SEs, the formation of amorphous $xLi_2O$-$HfCl_4$ solids was more difficult. Only $1.5Li_2O$-$HfCl_4$ exhibited relatively highest

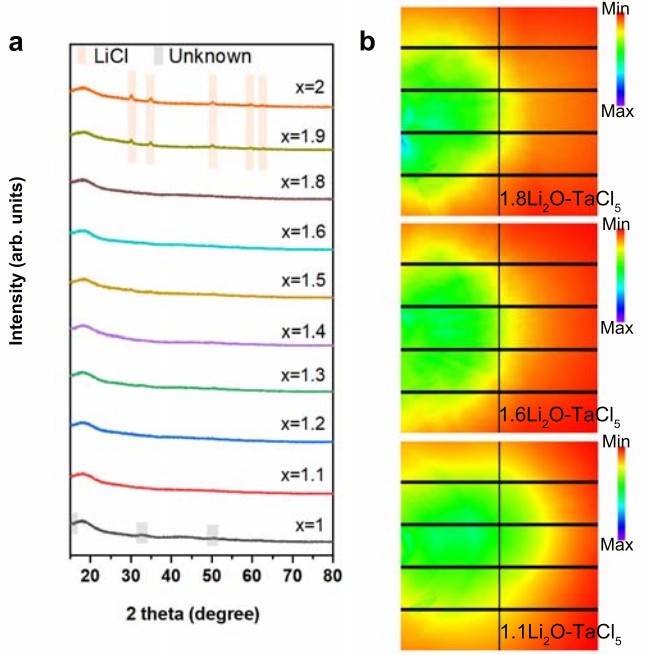
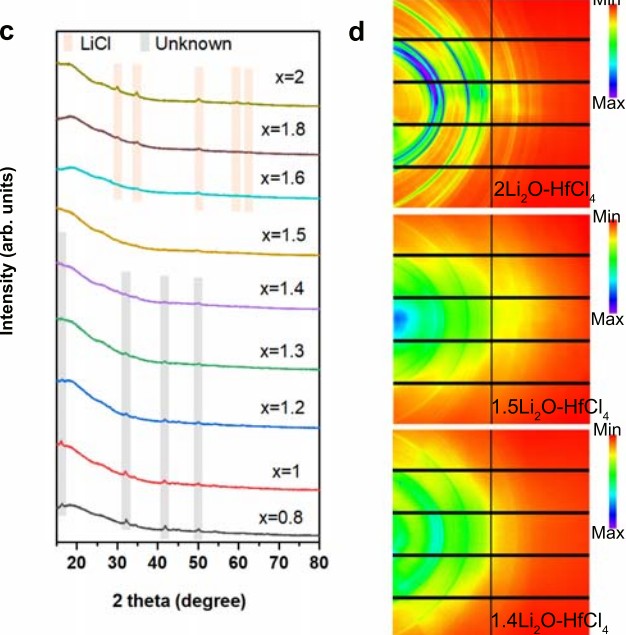

**Fig. 1 | Synthesis and identification of $xLi_2O$-$MCl_y$ (M = Ta or Hf) amorphous materials. a, c,** Lab-based XRD patterns for the as-prepared $xLi_2O$-$TaCl_5$ ($1 \leq x \leq 2$) (**a**) and $xLi_2O$-$HfCl_4$ ($0.8 \leq x \leq 2$) (**c**). The broaden peaks before 25° are the diffraction peaks of the Kapton films used to seal the tested powders to avoid any air exposure. **b, d** Synchrotron-based 2D diffraction images of $xLi_2O$-$TaCl_5$ (x = 1.8, 1.6, and 1.1) (**b**) and $xLi_2O$-$HfCl_4$ (x = 2, 1.5, and 1.4) (**d**). Source data are provided as a Source Data file.

amorphous content among all the prepared $xLi_2O\text{-}HfCl_4$ (Fig. 1d). For other $xLi_2O\text{-}HfCl_4$ compositions ($x \neq 1.5$), LiCl or unknown crystalline impurities could be easily observed.

## Ionic conductivity and Li-ion conduction analyses of the $xLi_2O\text{-}MCl_y$ materials

The ionic conductivities of the new amorphous SEs were determined by measuring the electrochemical impedance spectroscopy (EIS). The $xLi_2O\text{-}TaCl_5$ ($x = 1.0, 1.1, 1.2, 1.4, 1.6, 1.8, 1.9,$ and $2.0$) powders were cold-pressed into pellets for measurements. Their temperature-dependent EIS plots are shown in Supplementary Fig. 3. Figure 2a shows the extracted conductivity values at 25 °C. When the molar ratio of $Li_2O$ increased from 1 to 1.1, a surge of ionic conductivity (from $0.41 \times 10^{-3}$ to $5.3 \times 10^{-3}$ S cm$^{-1}$) could be observed. Remarkably, the $xLi_2O\text{-}TaCl_5$ ($1.1 \leq x \leq 1.8$) pellets retained similarly high ionic conductivities of around $6 \times 10^{-3}$ S cm$^{-1}$, which is at the top level among all other reported SEs (Supplementary Table 1). The highest ionic conductivity was $6.6 \times 10^{-3}$ S cm$^{-1}$ for an optimized composition of $1.6Li_2O\text{-}TaCl_5$. The activation energies determined from Arrhenius plots of

$xLi_2O\text{-}TaCl_5$ amorphous SEs showed low values from 0.241 eV to 0.277 eV (Fig. 2b and Supplementary Fig. 4). Meanwhile, Fig. 2c compares the ionic conductivities for $xLi_2O\text{-}HfCl_4$ system ($x = 0.8, 1, 1.2, 1.4, 1.5, 1.6, 1.8,$ and $2$) at 25 °C (see corresponding Nyquist plots in Supplementary Fig. 5). The $1.5Li_2O\text{-}HfCl_4$ SE in mostly amorphous state showed the highest ionic conductivity ($1.97 \times 10^{-3}$ S cm$^{-1}$) and a low activation energy (0.328 eV) among the $xLi_2O\text{-}HfCl_4$ series (Fig. 2d and Supplementary Fig. 6). Direct current (DC) measurements for the representative $1.6Li_2O\text{-}TaCl_5$ and $1.5Li_2O\text{-}HfCl_4$ amorphous SEs under ion-blocking and electron-blocking conditions[13,32] were also conducted as shown in Supplementary Fig. 7. The determined electronic conductivities were negligible (at $10^{-10}$ S cm$^{-1}$ order). The Li-ion conductivities calculated from the DC measurements agree well with the values we derived from the EIS measurements, confirming the $xLi_2O\text{-}MCl_y$ amorphous SEs as excellent Li-ion conductors.

In order to analyze the Li-ion mobility in $xLi_2O\text{-}MCl_y$ amorphous SEs, solid-state nuclear magnetic resonance (SSNMR) spectroscopy was adopted to provide the nuclide-specific information on structure and dynamics. First, qualitative information on Li-ion mobility[33,34] of

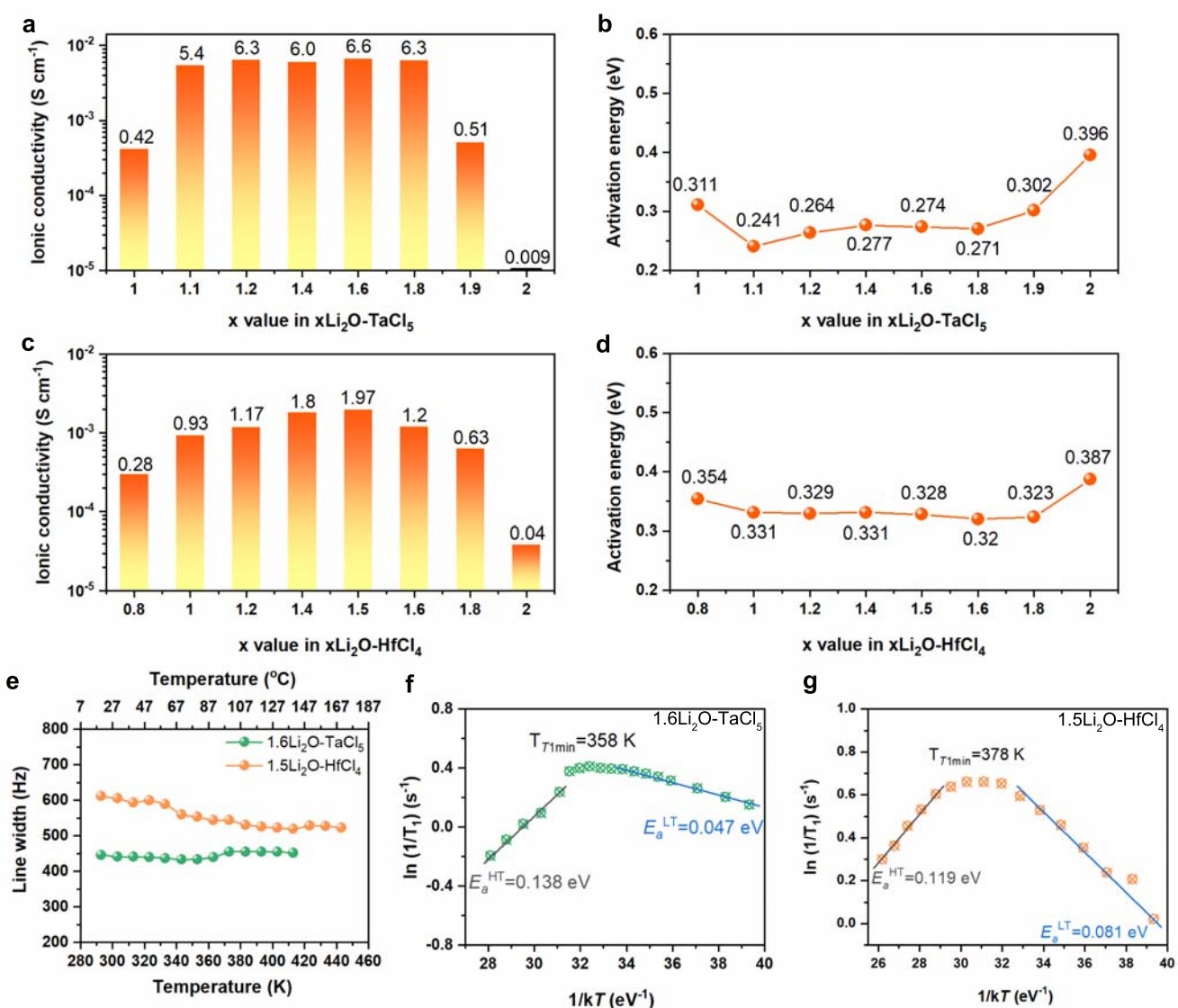

**Fig. 2 | Li-ion conductivity and Li-ion diffusion behaviors of $xLi_2O\text{-}TaCl_5$ and $xLi_2O\text{-}HfCl_4$ amorphous materials. a**, **c** The RT (25 °C) ionic conductivities for $xLi_2O\text{-}TaCl_5$ (**a**) and $xLi_2O\text{-}HfCl_4$ (**c**). **b**, **d** The corresponding activation energy values for $xLi_2O\text{-}TaCl_5$ (**b**) and $xLi_2O\text{-}HfCl_4$ (**d**) compounds. **e** The static $^7Li$ motional narrowing spectra of $1.6Li_2O\text{-}TaCl_5$ and $1.5Li_2O\text{-}HfCl_4$. **f**, **g** Temperature-dependent $^7Li$ SLR NMR rates for $1.6Li_2O\text{-}TaCl_5$ (**f**) and $1.5Li_2O\text{-}HfCl_4$ (**g**) measured in the laboratory frame of the reference. Source data are provided as a Source Data file.

the $xLi_2O$-$TaCl_5$ and $xLi_2O$-$HfCl_4$ systems was provided via $^7Li$ SSNMR temperature-dependent line-shape analyses. The selected $^7Li$ SSNMR spectra of the $1.6Li_2O$-$TaCl_5$ and $1.5Li_2O$-$HfCl_4$ samples in a temperature range of 293–443 K are displayed in Supplementary Fig. 8. For both samples, a strong resonance corresponding to the central transition (CT) of $^7Li$ ($I = 3/2$) was observed in each spectrum. The peaks due to the CT were narrow. The linewidths (Fig. 2e) were less than 1 kHz (i.e. around 450 Hz for $1.6Li_2O$-$TaCl_5$ and between 520–610 Hz for $1.5Li_2O$-$HfCl_4$) without significant variations in the temperature range of 293–443 K, implying that both systems were in the extreme narrowing regime. Such observation seemed to suggest a narrow distribution of slightly different Li-ion jumping rates and, by extension, a distribution of Li diffusion pathways, which is consistent with the amorphous nature of the materials.

Second, $^7Li$ NMR spin-lattice relaxation (SLR) time ($T_1$) allowed us to quantitatively determine the Li-ion jumping rates and activation energies corresponding to short-range as well as long-range ion diffusions in bulk electrolytes[35–37]. Fig. 2f, g show the plots of ln $(1/T_1)$ versus reciprocal temperature $(1/kT)$ for $1.6Li_2O$-$TaCl_5$ and $1.5Li_2O$-$HfCl_4$, respectively. A $1/T_1$ maximum was reached for each system, corresponding to 358 and 378 K for $1.6Li_2O$-$TaCl_5$ and for $1.5Li_2O$-$HfCl_4$, respectively. The fact that the peak maximum of $1.6Li_2O$-$TaCl_5$ appeared at a lower temperature than $1.5Li_2O$-$HfCl_4$ suggested higher Li-ion mobility in $1.6Li_2O$-$TaCl_5$[38]. Li-ion jump frequency can be deduced from the maximum condition ($\tau \cdot \omega_0 \approx 1$) at the relaxation rate peak[34,36,39,40]. Since the Larmor frequency of $^7Li$ at 9.4 T was 155.2 MHz, the Li-ion jump frequency ($1/\tau$) of $9.8 \times 10^8 s^{-1}$ occurred at 358 and 378 K for the $1.6Li_2O$-$TaCl_5$ and the $1.5Li_2O$-$HfCl_4$, respectively. The activation energies, $E_a^{HT}$ and $E_a^{LT}$, derived from the slopes of high-temperature (HT) and low-temperature (LT) flanks were not equal. Specifically, for the $1.6Li_2O$-$TaCl_5$ sample, the activation energies were $E_a^{HT} = 0.138$ eV and $E_a^{LT} = 0.047$ eV, whereas the $E_a^{HT} = 0.119$ eV and $E_a^{LT} = 0.081$ eV for $1.5Li_2O$-$HfCl_4$. Generally, the $E_a^{HT}$ and $E_a^{LT}$ are related by a relationship of $E_a^{LT} = (\beta - 1) E_a^{HT}$ where $1 < \beta \le 2$. For uncorrelated isotropic diffusion described by the BPP (Bloembergen, Purcell and Pound) model, $E_a^{HT}$ and $E_a^{LT}$ are equal, corresponding to $\beta = 2$[16]. In the present case, $\beta$ values were determined to be 1.34 and 1.68 for the $1.6Li_2O$-$TaCl_5$ and the $1.5Li_2O$-$HfCl_4$ samples, respectively, indicating structurally complex Li-ion conductions[40–42]. Generally, correlation effects (e.g., Coulomb interactions, correlated ion dynamics, structural disorders, etc.) are considered closely associated with the impacted Li-ion conduction[43,44]. In our studies of $^7Li$ SLR NMR for the amorphous SEs, the native structural disorder of the two amorphous samples was regarded as the major contributor towards the deviation of $\beta$ value off 2 ($1 < \beta < 2$), leading to smaller $E_a^{LT}$ values compared to those of $E_a^{HT}$[33,42]. Elaboration on the relevant correlation effects of locally disordered structure on the Li-ion migration is proposed as an interesting research direction that appeals to further attention.

## Local structure exploration of $xLi_2O$-$MCl_y$ amorphous SEs

The Li-ion transport environment and structural information of representative $xLi_2O$-$TaCl_5$ were investigated by using Raman spectroscopy, X-ray Photoelectron Spectroscopy (XPS) and XAS. Figure 3a depicts the Raman spectra of $xLi_2O$-$TaCl_5$ ($x = 1.2, 1.4, 1.6, 1.8,$ and 2). It was interesting to find that the bands at 180 $cm^{-1}$ and 406 $cm^{-1}$ for $xLi_2O$-$TaCl_5$ ($x = 1.2, 1.4, 1.6$ and 1.8) corresponded to Ta–Cl vibrations in trigonal bipyramidal $TaCl_5$[45,46], which implied a dissociation of $Ta_2Cl_{10}$ bi-octahedra to $TaCl_5$ trigonal bipyramid when introducing a moderate amount of $Li_2O$ into $TaCl_5$ under high-energy ball-milling conditions. In comparison with noticeable Ta–Cl features, Ta–O fingerprints in $xLi_2O$-$TaCl_5$ were broader to show double/triple-coordinated oxygen stretching (O-3Ta/O-2Ta) vibrations[47,48]. This was a key signal that O atoms mainly act as bridges to connect Ta-centered trigonal bipyramids in $xLi_2O$-$TaCl_5$ amorphous SEs. Such an observation

could also be verified in O 1$s$ XPS spectra in Fig. 3b, in which bridging oxygens with a binding energy of 532.1 eV[49,50] showed increased fraction along with the growth of the feeding $Li_2O$. This was consistent with the previous reports for oxysulfide amorphous SEs, which indicated oxygens showed strong amorphous formation ability to preferentially become bridging oxygens to connect polyhedra in a short range[51].

Since oxygens mainly contributed to form the structure of amorphous SEs, it was highly possible that chlorines took the responsibility to conduct Li ions. The chloride anion chemistry has been proved being beneficial for Li-ion migration because of a relatively large anion radius, large anion polarizability, and weak interaction with Li-ion[52]. Because of that, unsaturated Ta–Cl···Li bonds were proposed in $xLi_2O$-$TaCl_5$ amorphous SEs (Fig. 3c). The interactions between lithium and chlorine could be proved by Cl $K$-edge XAS. As shown in Fig. 3d, despite a pre-edge feature P1 (2822.3 eV) was indistinctly found in $xLi_2O$-$TaCl_5$ amorphous SEs because a mixing of Cl $p$-orbitals and Ta $d$-orbitals increased covalency[53], the near-edge spectra of $xLi_2O$-$TaCl_5$ amorphous SEs (P2, P3, P4 and P5) were very similar to that of the LiCl. These peaks reflected the Cl 1$s$ electron transition process to unoccupied states and multiple scatterings, giving direct proof that Li atom was the nearest neighbor around Cl atom. Besides, it was interesting to find the oscillation in the extended range (from 2843 to 2920 eV) of each $xLi_2O$-$TaCl_5$ spectrum was identical with that of the LiCl, while the $xLi_2O$-$TaCl_5$ curves even showed a stronger amplitude than the LiCl one. This meant the Li-ion mobility environment in $xLi_2O$-$TaCl_5$ amorphous SEs was similar to that in LiCl but with rich Li···Cl or Cl···Li···Cl interactions.

Then, the Ta $L_3$-edge extended X-ray absorption fine structure (EXAFS) spectra were employed to determine the atomic-scale chemical environment of amorphous $xLi_2O$-$TaCl_5$ SEs. In the Ta $L_3$-edge XAS spectra, since the white line (WL) at Ta $L_3$-edge corresponds to the dipolar transition from $2p_{3/2}$ core levels to unoccupied Ta $5d$ states, the WL intensity and peak position increase when the oxidation state of Ta increases, and vice versa[54]. As depicted in Fig. 4a, the XANES spectra of $xLi_2O$-$TaCl_5$ ($x = 1.2, 1.4, 1.6, 1.8$) possessed a similar feature (9888 to 9898 eV) with that of the $TaCl_5$. The absorption edge energy ($E_0$) at Ta $L_3$-edge for $xLi_2O$-$TaCl_5$ was between 9883 eV ($Ta_2O_5$) and 9882 eV ($TaCl_5$), ascribing to the oxidation of $TaCl_5$ when $Li_2O$ was added. Figure 4b shows Ta $L_3$-edge EXAFS of $xLi_2O$-$TaCl_5$ in $k$-space. The weakened amplitude in the range of 4 to 8 $Å^{-1}$ and a low-$k$ phase shift could be observed especially for $1.6Li_2O$-$TaCl_5$ and $1.8Li_2O$-$TaCl_5$. This was a signal for elongation and disorder of Ta–Cl bonds caused by Ta–O bonding, which could also be reflected in the XPS results (Fig. 3c). To further resolve the coordination of Ta in $xLi_2O$-$TaCl_5$, phase-uncorrected radial distribution functions (RDF) after Fourier Transformed (FT) EXAFS and wavelet transformed (WT) EXAFS were conducted[55,56]. Based on the Ta–O and Ta–Cl scattering paths in referential $TaCl_5$ and $Ta_2O_5$ (Supplementary Fig. 9), Ta in each $xLi_2O$-$TaCl_5$ amorphous SE could be recognized to be coordinated by O and Cl (Fig. 4c–e, j, h). Intensive Ta-O signals could be observed for $1.6Li_2O$-$TaCl_5$ and $1.8Li_2O$-$TaCl_5$. EXAFS fitting (Supplementary Fig. 10) provided a semi-quantitative explanation about these differences in $xLi_2O$-$TaCl_5$ amorphous SEs, and the results were listed in Supplementary Table 2. The coordination number (CN) nearest to Ta could be estimated to be 5 in each $xLi_2O$-$TaCl_5$ compound, consistent with the coordination situations for Ta-centered trigonal bipyramids. As a result, we could determine that the local structure in superionic $1.2Li_2O$-$TaCl_5$ and $1.4Li_2O$-$TaCl_5$ amorphous SEs was mainly $[TaCl_4O]^-$ trigonal bipyramid (Fig. 4f). Feeding more $Li_2O$ induced mixed short-order structures as $[TaCl_4O]^-$ and O-rich $[TaCl_{5-a}O_a]^{a-}$ ($2 \le a < 5$) in $1.6Li_2O$-$TaCl_5$ and $1.8Li_2O$-$TaCl_5$ amorphous SEs (Fig. 4i). Besides, the connection of local structures in $xLi_2O$-$TaCl_5$ amorphous SEs could also be found as noticeable Ta–O–Ta (also write as Ta–Ta) resonances (12 $Å^{-1}$) in Supplementary Fig. 11, which was of highly similarity with the Ta–O–Ta resonance in $Ta_2O_5$. Therefore, $[TaCl_4O]^-$ and $[TaCl_{5-a}O_a]^{a-}$

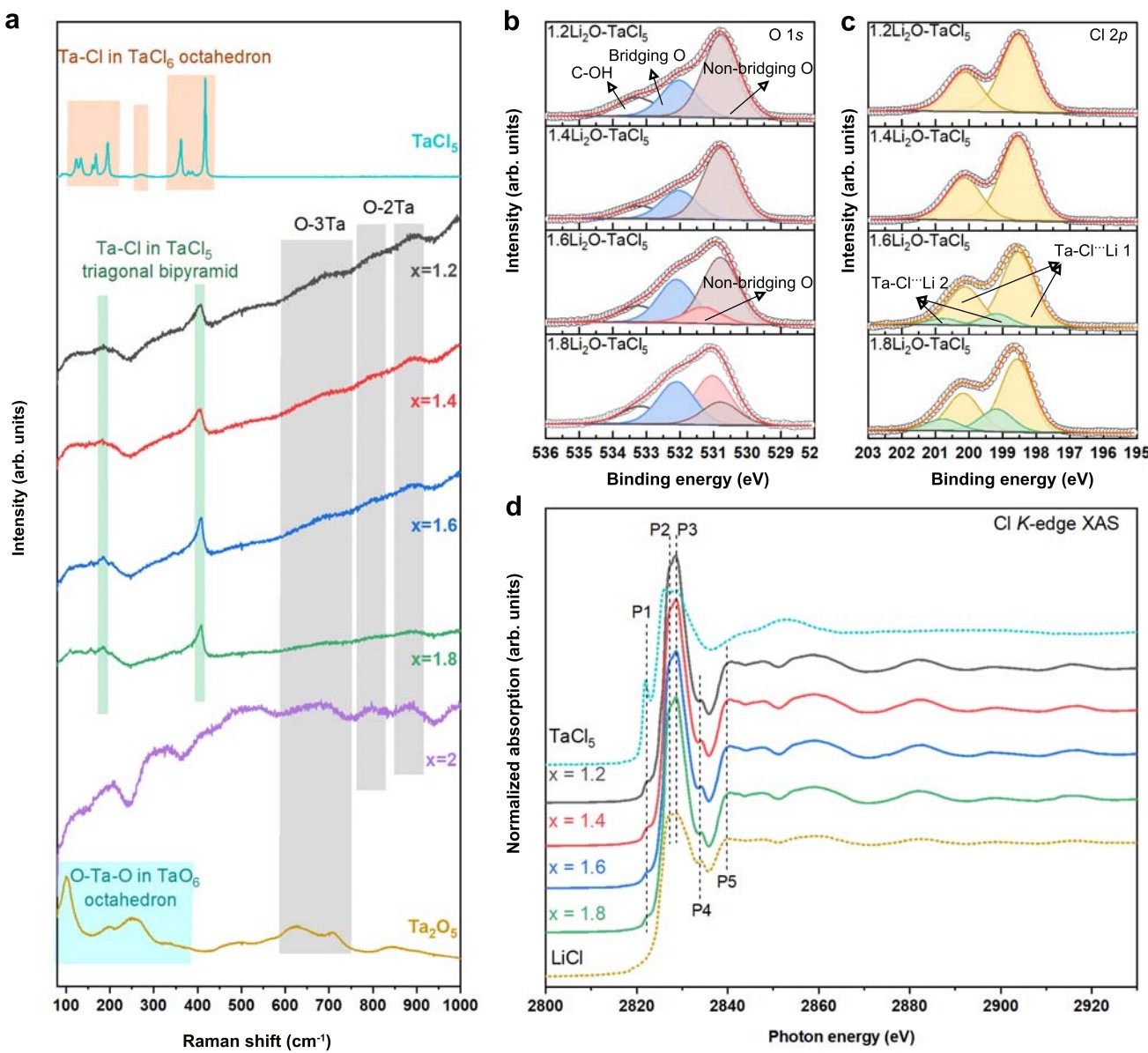

**Fig. 3 | Structure characterizations and mix anion functions of xLi$_2$O·TaCl$_5$ amorphous SEs. a** Raman spectra of the xLi$_2$O·TaCl$_5$ compounds, Ta$_2$O$_5$, and TaCl$_5$. **b–d** O 1$s$ XPS (**b**), Cl 2$p$ XPS (**c**), and Cl $K$-edge XAS (**d**) for xLi$_2$O·TaCl$_5$ amorphous SEs. Source data are provided as a Source Data file.

shared most of the oxygens to form the networks of amorphous SEs were further confirmed.

Based on above structural information, explanations for high conductivity of xLi$_2$O·TaCl$_5$ amorphous SEs could be proposed as following. First, disordered and irregular [TaCl$_{5-a}$O$_a$]$^{a-}$ (1 ≤ a < 5) arrangements in amorphous xLi$_2$O·TaCl$_5$ SEs made it possible to form rich Li−Cl interactions and distorted Li−Cl sublattices. Second, corner-shared oxygen (O-2Ta) networks could induce a much wider range of distortions in Li sites[57]. The distorted lithium sites in O-2Ta networks were the predominance to realize a Li-ion energy landscape with low migration energy. At the same time, oxygens at the bridging positions would enlarge the doorway radius for easy access of Li ions[58]. Third, the unsaturated Ta−Cl···Li bonds in [TaCl$_{5-a}$O$_a$]$^{a-}$ (1 ≤ a < 5) showed weak Coulombic forces between lithium and chlorine, making it easier for Li ions escape from one site and jump to another. In short, oxygen incorporation is beneficial to the amorphization of xLi$_2$O·TaCl$_5$. The induced disordered structures in amorphous xLi$_2$O·TaCl$_5$ lead to a sharply increased ionic conductivity and decreased activation energy compared to the single-anion Li-Ta-Cl sample (see comparison data in Supplementary Fig. 12).

The local structures of amorphous 1.5Li$_2$O·HfCl$_4$ were also explored by Hf $L_3$-edge XANES and EXAFS. Similar to the behaviors in xLi$_2$O·TaCl$_5$, the E$_0$ of Hf $L_3$-edge XANES in 1.5Li$_2$O·HfCl$_4$ was between those of the in HfCl$_4$ and HfO$_2$ (Supplementary Fig. 13). Combing with the WT-EXAFS spectra (Supplementary Fig. 14), it was shown that in 1.5Li$_2$O·HfCl$_4$, Hf was also nearest coordinated by O and Cl (Supplementary Fig. 15 and Supplementary Table 3). The XAS of superionic conductive 1.5Li$_2$O·HfCl$_4$ amorphous SE and poor ionic conductor 2Li$_2$O·HfCl$_4$ were compared, which showed 1.5Li$_2$O·HfCl$_4$ amorphous SE with more Hf−Cl bonds while 2Li$_2$O·HfCl$_4$ with obvious Hf−O interactions (Supplementary Fig. 13c). Based on the structural information in xLi$_2$O·TaCl$_5$, it was reasonable to know that the formation the abundant terminal chlorines and moderate bridging oxygens in 1.5Li$_2$O·HfCl$_4$ amorphous SEs was predominant to the fast Li-ion conduction.

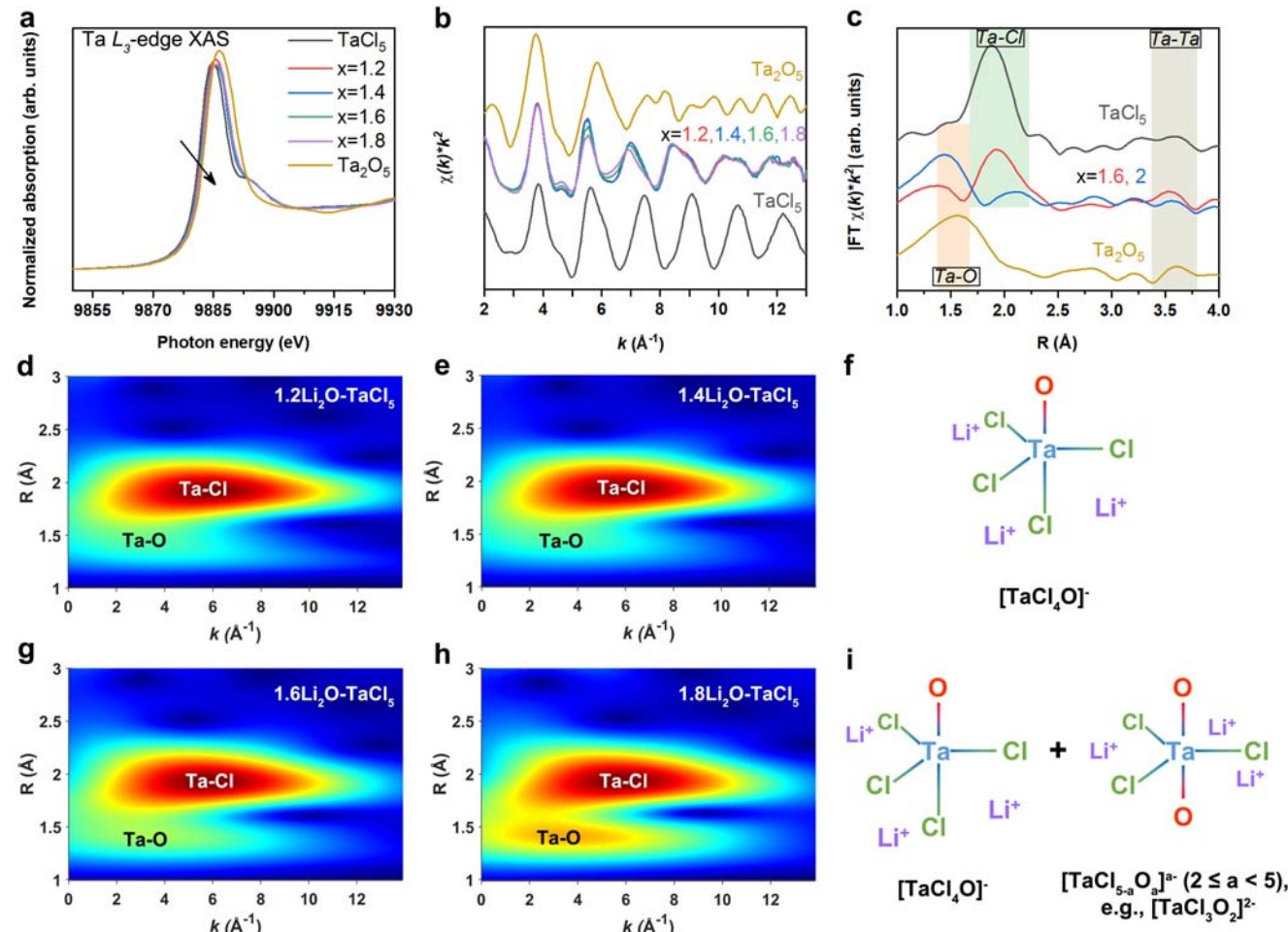

**Fig. 4 | Local structure analyses of xLi₂O-TaCl₅ amorphous SEs. a–c** XANES (**a**), EXAFS (**b**), and FT-EXAFS (**c**) for xLi₂O-TaCl₅ amorphous SEs, as well as Ta₂O₅ and TaCl₅ at Ta $L_3$-edge. **d, e, g, h** WT spectra of 1.2Li₂O-TaCl₅ (**d**), 1.4Li₂O-TaCl₅ (**e**), 1.6Li₂O-TaCl₅ (**g**), and 1.8Li₂O-TaCl₅ (**h**) at Ta $L_3$-edge with a $k^2$ weighting. **f, i** The

schematics of the local structures in superionic xLi₂O-TaCl₅ amorphous SEs (x = 1.2 and 1.4 in (**f**); x = 1.6 and 1.8 in (**i**)). (The local structures mainly display the coordination and possible geometry of xLi₂O-TaCl₅ amorphous SEs. Bond angles and Li-ion numbers are not accurate.) Source data are provided as a Source Data file.

## Electrochemical performance of ASSBs

As promising SEs for bulk-type all-solid-state batteries (ASSBs), the electrochemical stability windows of 1.6Li₂O-TaCl₅ and 1.5Li₂O-HfCl₄ amorphous SEs were measured (Supplementary Fig. 16). Although the amorphous SEs were thermodynamically unstable against metal anodes (such as Li and Li-In) due to the reduction of Ta⁵⁺ or Hf⁴⁺, they showed good oxidative stability and are promising to tolerate cathode active materials beyond 4 V. The electrochemical performances of 1.6Li₂O-TaCl₅ and 1.5Li₂O-HfCl₄ amorphous SEs were then evaluated with LiNi₀.₈₃Co₀.₁₁Mn₀.₀₆O₂ (NCM83) and LiCoO₂ (LCO). The NCM83 ASSB with 1.6Li₂O-TaCl₅ SE exhibited good rate performance as shown in Fig. 5a. The charge-discharge curves depict high reversible capacities of 189.5 mAh g⁻¹, 177.1 mAh g⁻¹, 161.7 mAh g⁻¹, 138 mAh g⁻¹, 105.7 mAh g⁻¹ and 83 mAh g⁻¹ at 0.1 C, 0.2 C, 0.5 C, 1 C, 2 C, and 3 C, respectively (1 C = 200 mA g⁻¹). As for the NCM83 ASSB with 1.5Li₂O-HfCl₄ SE, the rate performance was comparable with the 1.6Li₂O-TaCl₅ cell at lower rates (191.1 mAh g⁻¹ @ 0.1 C, 171.9 mAh g⁻¹ @ 0.2 C, 149.2 mAh g⁻¹ @ 0.5 C) (Fig. 5b). NCM83 ASSBs integrated with either amorphous SE showed superior long cycle life at designated specific currents. As shown in Fig. 5c, the ASSB with 1.6Li₂O-TaCl₅ SE demonstrated highly stable cycling performance for over 300 cycles with a capacity retention of 92.9% at 1 C. The 1.5Li₂O-HfCl₄ ASSB also retained 89.6% of its initial reversible capacity after 300 cycles at 0.5 C (Fig. 5d). In particular, the impressive cycling durability at 2 C was demonstrated for the solid cell after evaluation of its rate capability

(Fig. 5a). As depicted in Fig. 5e, there was a capacity retention of 90.7% after 2400 cycles. The charge-discharge curves suggested effective electrochemical reversibility during the long-cycling measurement (Supplementary Fig. 17). Due to the high ionic conductivity of 1.6Li₂O-TaCl₅, we further examined the ASSBs at a low temperature of −10 °C. The LCO ASSB exhibited an initial reversible capacity of 100.6 mAh g⁻¹ with a Coulombic efficiency (CE) of 94.9%. The cell maintained a high capacity of 93.7 mAh g⁻¹ after 100 cycles and stably operated for over 300 cycles with a capacity retention of 79.2% (average CE: 99.94%) (Fig. 5f). Meanwhile, despite that the 1.5Li₂O-HfCl₄ SE possessed a lower ionic conductivity of 3.4 × 10⁻⁴ S cm⁻¹ at −10 °C, the LCO solid cell with 1.5Li₂O-HfCl₄ could still stably cycle over 300 cycles at 0.2 C (1 C = 140 mA g⁻¹) (Fig. 5g). The rate performance of LCO ASSBs at −10 °C was also decent with either Ta-based or Hf-based amorphous SE (Supplementary Fig. 18). Finally, in order to demonstrate the practical prospect of amorphous SEs, 1.6Li₂O-TaCl₅ was chosen to be applied in pouch cell, which exhibited decent cycling performance in Supplementary Fig. 19.

In summary, we report a series of new amorphous superionic conductors, xLi₂O-TaCl₅ (x = 1.1–1.8) and xLi₂O-HfCl₄ (x = 1.5), which can be prepared via a one-step ball-milling method. Among them, the optimized 1.6Li₂O-TaCl₅ and 1.5Li₂O-HfCl₄ amorphous SEs possess high ionic conductivities of 6.6 × 10⁻³ S cm⁻¹ and 1.97 × 10⁻³ S cm⁻¹, respectively, at 25 °C. The local environment in the representative superionic Ta-based amorphous SEs is identified as [TaCl₅₋ₐOₐ]ᵃ⁻

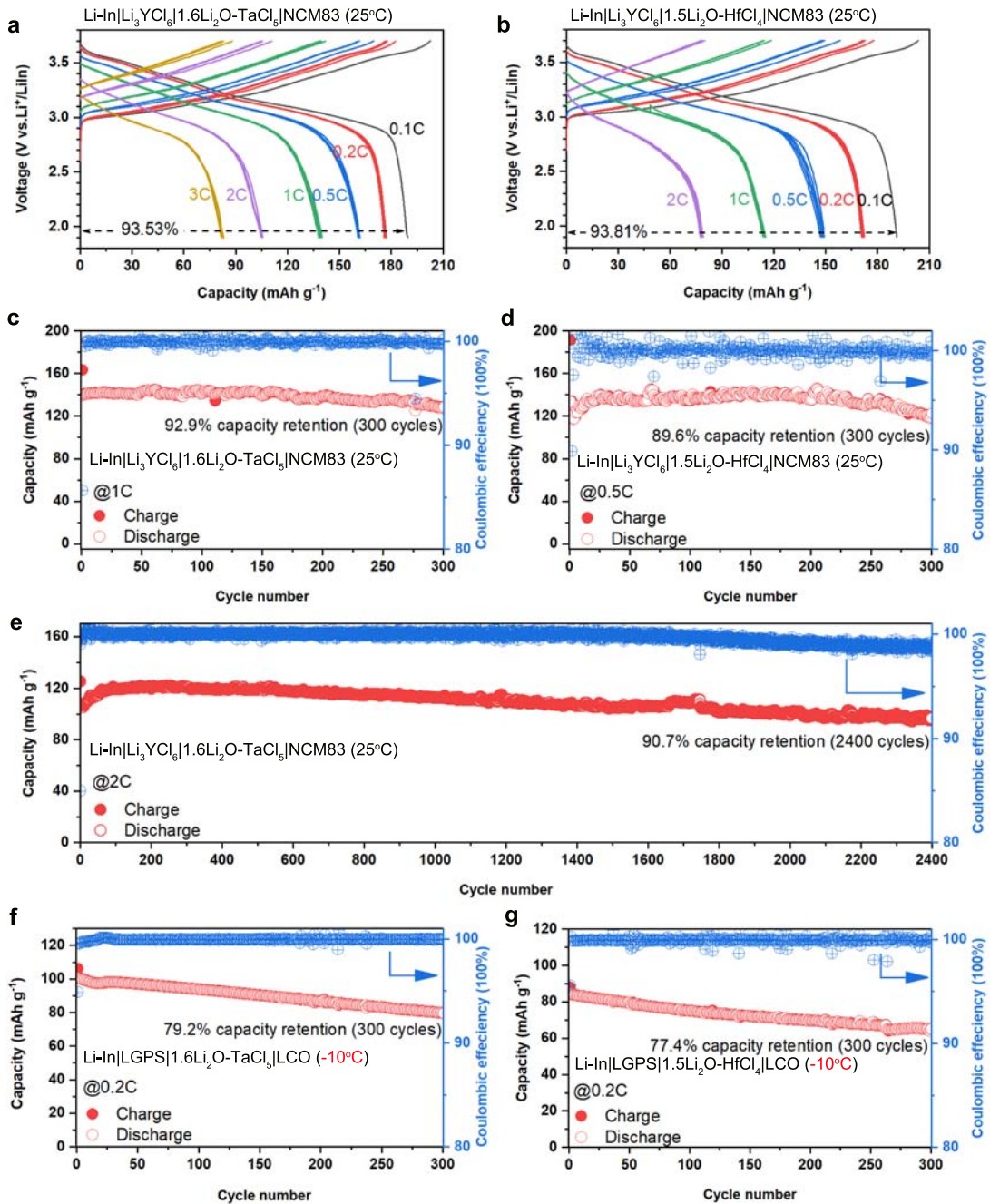

**Fig. 5 | Electrochemical performances of ASSBs using 1.6Li$_2$O-TaCl$_5$ or 1.5Li$_2$O-HfCl$_4$ amorphous SEs. a, b** Charge–discharge curves extracted from rate performance measurements of NCM83 ASSBs at room temperature (RT, 25 °C) with 1.6Li$_2$O-TaCl$_5$ (**a**) and 1.5Li$_2$O-HfCl$_4$ (**b**). **c–e** RT cycling performances of NCM83 ASSBs with 1.6Li$_2$O-TaCl$_5$ (**c**) and 1.5Li$_2$O-HfCl$_4$ (**d**), and long-life cycling of NCM83 ASSB using 1.6Li$_2$O-TaCl$_5$ at 2 C (**e**). **f, g** −10 °C cycling performances of LCO ASSBs using 1.6Li$_2$O-TaCl$_5$ (**f**) and 1.5Li$_2$O-HfCl$_4$ (**g**). Note: The specific currents of 1 C for NCM83 and LCO cathode materials are corresponding to 200 mA g$^{-1}$ and 140 mA g$^{-1}$, respectively. Source data are provided as a Source Data file.

($1 \le a < 5$) trigonal bipyramids in which abundant terminal chlorines directly interact with Li ions weakly. Bridging oxygens that serve as joints in the networks of amorphous SEs can induce a wide range of distortions in Li sites. Fast Li-ion conduction in xLi$_2$O-TaCl$_5$ amorphous SEs benefits from such a mixed anion chemistry. The 1.6Li$_2$O-TaCl$_5$ and 1.5Li$_2$O-HfCl$_4$ amorphous SEs also show good cathode compatibility with conventional layered oxide cathode materials (NCM83 and LCO), performing outstanding electrochemical performances at both 25 °C and −10 °C. This study shall provide insights into the Li-ion dynamics

and design principles of amorphous SEs, leading to a key advancement for ASSBs.

## Methods

### Synthesis of xLi$_2$O-MCl$_y$ (M = Ta or Hf) solid electrolytes

TaCl$_5$ (Sigma Aldrich, 99.99%), HfCl$_4$ (Sigma Aldrich, 98%), LiCl (Sigma Aldrich, reagent grade), and Li$_2$O (Alfa Aesar, 99.5%) were used as the raw materials. The starting materials for each compound were mixed in an argon-filled glovebox (H$_2$O < 0.1 ppm, O$_2$ < 0.1 ppm).

The resulting mixture (1 g) was then placed in a zirconia ball milling pot along with 40 g zirconia balls. Low-speed ball milling (100 rpm for 2 h) was first run to ensure all the precursors mixed well, followed by a high-speed ball milling process of 500 rpm for 10 h. Next, the ball-milled products were transferred into the glovebox for further use. Similarly, the $Li_3YCl_6$ electrolyte was prepared by ball milling $YCl_3$ and LiCl in mole ratio of 1: 3. The starting materials (1 g) were weighted and pre-mixed in an argon-filled glovebox, which then placed in a zirconia ball milling pot along with 40 g zirconia balls. Low-speed ball milling (100 rpm for 2 h) was first run to ensure all the precursors mixed well, followed by a high-speed ball milling process of 500 rpm for 10 h.

### Measurements of ionic and electronic conductivities

Ionic conductivity of as-prepared SEs was evaluated using electrochemical impedance spectroscopy (EIS) with two stainless steel rods as blocking electrodes. The SE powders (100-150 mg) were cold-pressed into pellets under ~300 MPa. The thickness of the pellets was between 0.04 cm and 0.06 cm. EIS measurements were performed using a multichannel potentiostat 3/Z (German VMP3). The applied frequency range was 1 Hz–7 MHz and the voltage amplitude was 20 mV. The temperature control was realized in an ESPEC Environmental Test Chamber. The cell assembly process for DC measurements was similar with that for EIS tests. To determine the electronic conductivity, the current responses of the cell was measured at a range of constant voltages for 60 min each. The applied voltage ranged from 0.1 to 0.5 V with a step size of 0.1 V. The DC Li-ion conductivity was evaluated with a symmetric cell configuration of $Li|Li_6PS_5Cl|xLi_2O\text{-}MCl_y|Li_6PS_5Cl|Li$ under a bias voltage for 30 min. The bias voltage was applied at 5, 10, 15, 20, and 25 mV consecutively. The $Li_6PS_5Cl$ SE (provided by China Automotive Battery Research Institute Co, Ltd) was used to prevent direct contact between Li metal and $xLi_2O\text{-}MCl_y$ SE.

### Linear sweep voltammetry (LSV) test

Approximately 80 mg of the $1.6Li_2O\text{-}TaCl_5$ or $1.5Li_2O\text{-}HfCl_4$ amorphous SE powder was cold-pressed into a pellet. A 10-mg mixture of amorphous SE and carbon black (CB) (8:2 wt./wt.) was uniformly covered on one side of the pellet as working electrode. Li foil was attached on the other side of the pellet as both counter and reference electrode. A $Li_6PS_5Cl$ interlayer (~40 mg) was adopted to avoid the incompatibility between amorphous SE and metallic Li. The LSV measurements were conducted using a versatile multichannel potentiostat 3/Z (VMP3) with a positive scan range from open-circuit voltage (OCV) to 6 V and a negative scan range from OCV to 0 V. The scan rate was 0.1 mV s$^{-1}$.

### Assembly and electrochemical characterizations of ASSBs

60 mg of $1.6Li_2O\text{-}TaCl_5$ and $1.5Li_2O\text{-}HfCl_4$ amorphous SEs were pressed at ~300 MPa to form a solid electrolyte layer (10 mm diameter), respectively. 10 mg of amorphous SE/NCM83 composite (3:7 mass ratio) was uniformly spread onto the surface of the one side of electrolyte layer and pressed with ~360 MPa for 5 min. NCM83 ($LiNi_{0.83}Co_{0.11}Mn_{0.06}O_2$) cathode material (polycrystalline particle size: ~3 μm) was provided by China Automotive Battery Research Institute Co, Ltd. Subsequently, Li-In alloy was placed on the other side of the electrolyte layer and pressed by ~120 MPa for 3 min. The Li-In alloy was prepared by pressing a piece of In foil (ϕ 10 mm, thickness 0.1 mm) and a piece of Li foil (ϕ 10 mm, thickness 20 μm) together under ~60 MPa for 5 min. To prevent the direct contact between amorphous SE and Li-In. 40 mg of $Li_3YCl_6$ powder pressed into pellet was served as the interlayer at the anode side. The obtained internal pellet cell was sandwiched between two stainless-steel rods as current collectors. Finally, a stack pressure of ~80 MPa was applied to the solid cell for various electrochemical tests. All cell fabrication processes were carried out in an Ar-filled glove box ($H_2O$, $O_2 < 0.1$ ppm). For the −10 °C full-cell test, the assemble process was similar as above. However,

we changed the anode electrolyte to $Li_{10}GeP_2S_{12}$ (LGPS) to provide a high ionic conductivity at −10 °C. LGPS was purchased from MSE Supplies LLC, showing a high ionic conductivity around $6 \times 10^{-3}$ S cm$^{-1}$ at 25 °C with cold-pressed pellet. LCO was used as cathode material. The specific currents of 1 C for NCM83 and LCO cathode materials are corresponding to 200 mA g$^{-1}$ and 140 mA g$^{-1}$, respectively. The electrochemical performances were evaluated using the Neware and Land battery testing system. The temperature of 25 °C (RT) for the battery testing was realized in a designated battery testing lab equipped with a temperature control system. The battery tests under the temperature of −10 °C were realized by a freezer manufactured by Thermo Fisher Scientific. Prior to the tests, all cells were rest and equilibrated for 12 h to reach target temperatures.

### Fabrication of all-solid-state pouch cell

The all-solid-state pouch cell was fabricated by stacking layers of NCM83/$1.6Li_2O\text{-}TaCl_5$ cathode, SE separators ($Li_3YCl_6$ and $1.6Li_2O\text{-}TaCl_5$), and Li-In alloy anode. The membranes of cathode composites and SEs were made by dry-film processing method[59], where 0.5 wt% Polytetrafluoroethylene (PTFE) were added to induce the formation of doughs and followed by calendaring to the target thickness (~80 um). The loading of the cathode was 13.125 mg cm$^{-2}$. Stacking each layers was completed in the Ar-filled glovebox, which was then sealed in plastic vacuum bag for transferring to a dry room for further packing in the aluminum-plastic bag. A pressure of ~10 MPa was applied on the pouch cell during the cycling performance test using Neware battery testing system.

### Characterization methods

Lab-based XRD measurements were performed on Bruker AXS D8 Advance with Cu Kα radiation (λ = 1.5406 Å). Kapton tape was used to cover the sample holder to prevent from the air exposure.

Raman spectra were measured with a HORIBA Scientific LabRAM HR Raman spectrometer operated under laser beam at 532 nm. Electrolyte powders were attached on a carbon tape and covered by a transparent cover glass for the test.

$^7$Li SSNMR SLR measurements were performed on a Varian Infinity Plus wide-bore NMR spectrometer equipped with an Oxford wide-bore magnet ($\mathbf{B}_0 = 9.4$ T). The $^7$Li Larmor frequency was 155.248 MHz. The π/2 and π pulse length were determined to be 2.3 and 4.5 μs, respectively. Chemical shifts were referenced with respect to a 1.0 M LiCl solution. The electrolyte sample was sealed in custom-made Teflon tubes (ϕ = 4.7 mm) in an argon-filled glovebox ($H_2O < 0.1$ ppm, $O_2 < 0.1$ ppm). The $^7$Li spin-lattice relaxation times ($T_1$) at different temperatures were determined using an inversion-recovery NMR experiment. The testing temperature ranges from 293–443 K.

Ta $L_3$-edge and Hf $L_3$-edge XAS data were collected at the 44 A beamline of Taiwan Photon Source (TPS) of the National Synchrotron Radiation Research Center (NSRRC) in Taiwan. The spectra were recorded in transmission mode. Cl K-edge XAS (FY mode) were collected at the SXRMB beamline at Canadian Light Source (CLS). The above data were processed with Athena and Artemis softwares. Synchrotron-based 2D XRD images were collected at VESPERS beamline at CLS. The 2D diffraction data were recorded on a Pilatus 1 M detector with a photon energy of 13 keV (λ = 0.9537 Å). Profex and ALBULA softwares were used to process the data.

XPS were collected at Surface Science Western (SSW) in Canada by using Krotos AXIS Ultra Spectrometer system. Monochromatic Al K(alpha) source was adopted. There was a specially designed inert transfer vessel allowing for SEs samples in a glove box, and transferring to the instrument without air exposure. High resolution analyses were carried out with an analysis area of 100 microns and a pass energy of 40 eV.

## Data availability

All data that support the findings of this study are provided within the paper and its Supplementary Information. All additional information is available from the corresponding authors upon request. Source data are provided with this paper.

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

## Acknowledgements

This research was supported by the Natural Sciences and Engineering Research Council of Canada (NSERC), the Canada Research Chair Program (CRC), the Canada Foundation for Innovation (CFI), Ontario Research Foundation (ORF), and the University of Western Ontario (UWO). The synchrotron research was performed at the Canadian Light Source, a national research facility of the University of Saskatchewan, which was supported by the CFI, NSERC, the National Research Council (NRC), the Canadian Institutes of Health Research (CIHR), the Government of Saskatchewan, and the University of Saskatchewan.

## Author contributions

S.Z., F.Z., and X.S. conceived and designed the experiments. S.Z. prepared samples and carried out the main experiments. S.Z. and F.Z. examined the electrochemical performance. S.Z. analyzed the XAS data and discussed with J.C. and T.S. S.A. and Y.H. contributed to the NMR characterizations and data analysis. J.F., J.L., L.C., R.F., and M.S. carried out the synchrotron-related tests. J. Liang and X.L. helped with the Raman measurements and analysis. Y.Z helped with the SEM measurements. S.Z. and F.Z. wrote and revised the paper. All the authors commented on the manuscript.

## Competing interests

The authors declare no competing interests.
