## [Peer Review File · Nature Communications]

REVIEWER COMMENTS

Reviewer #1 (Remarks to the Author):

The authors talk about the synthesis, characterization, and test of new glassy electrolytes of $x\text{Li}_2\text{O}-\text{TaCl}_5$ and $x\text{Li}_2\text{O}-\text{HfCl}_4$. The structure is well characterized, the Li-ion conductivity is superior, and the battery performance is attractive. I thus recommend the manuscript to Nature Comm. upon addressing the following questions:

- 1) The operational stack pressure should be mentioned for all batteries. Only the fabrication pressures are mentioned in Method for now.
- 2) It's mentioned LCO is used in Method, but all batteries I see are by NMC83.
- 3) More details should be provided for NMC83, purchased? SC v.s. PC?
- 4) It's nice to see a pouch cell in SI. But more details should be provided, as it's not mentioned in Method, i.e., cathode loading, separator thickness, slurry casting?, layer by layer?, operational pressure, sealing, etc.
- 5) It's interesting to see that E_{a_LT} is smaller than E_{a_HT} , and the authors mentioned structure disorder as the contributor. Is the effect static or dynamic (e.g. "Advanced Materials, 2207411 (2022))? A brief discussion of the mechanism along this direction will further benefit the readers.

Reviewer #2 (Remarks to the Author):

The composition of the present study is, for example, $\text{Li}_{3.2}\text{TaCl}_5\text{O}_{1.6}$. Thus, this material can be regarded as a series of oxide-ion-doped chloride-based solid electrolytes. In such a sense, how about the conductivity of the material in the $\text{LiCl}-\text{TaCl}_5$ system? By comparing the behavior of the $\text{LiCl}-\text{TaCl}_5$ system, the effect of oxygen incorporation can be discussed.

How the authors determined that the observed conductivity is "ionic conductivity"? The author should show the transport number of electronic conduction, ionic conduction, and Li-ion conduction.

In the Figures for charge-discharge curves, the cell configuration should be shown, e.g. $\text{Li-In/Li}_3\text{YCl}_6/x\text{Li}_2\text{O}-\text{TaCl}_5/\text{LiCoO}_2$.

In the low-temperature charge-discharge test, the authors described that LGPS was used. The cell configuration was $\text{Li-In/LGPS}/x\text{Li}_2\text{O}-\text{TaCl}_5/\text{LiCoO}_2$?

Anyway, the electrochemical window of the present solid electrolytes should be discussed.

The authors use the expression "glassy". However, the authors only show that the materials are amorphous by XRD pattern and no evidence for a "glassy state." In the strict sense, "glass" and "amorphous" is not the same category of material, and "glassy" material should show the glass transition behavior. The reviewer strongly recommends using "amorphous" in the present study.

Response to the Reviewers' Comments (Manuscript NCOMMS-22-47305-T)

We are very grateful for the reviewers' constructive comments and insightful suggestions for improving the quality of this manuscript. During the revision, we have carried out additional experiments and literature review to gain a better understanding of the oxychloride amorphous solid electrolytes. As a result, we have updated *1 figure* in the revised manuscript. More convincing discussions, experiment details, and references have also been added in the appropriate parts of the revised manuscript. Besides, *3 figures* have been added and *1 figure* have been updated in the revised supplementary information.

The major experiments and updates include:

1. In response to the suggestions of Reviewer #1, we have supplied the operational stack pressure for all batteries, detailed information for NCM83, and the fabrication details for all-solid-state pouch cell. We have also highlighted the LCO all-solid-state batteries by providing cell configurations or bolding the text. Additionally, we have updated the discussions towards the structure disorder and $E_{a_LT} < E_{a_HT}$ by reviewing more references.
2. In addressing the comments of Reviewer #2, we have elucidated the effects of oxygen incorporation through the synthesis of Li-Ta-Cl materials and characterization of their phase compositions and ionic conductivities. We have also conducted DC measurements under ion-blocking and electron-blocking conditions to identify the electronic conductivity and Li-ion conductivity of our reported oxychloride solid electrolytes. The electrochemical stability windows were provided through doing the LSV test. Moreover, after checking relevant references, we have determined that our reported solid electrolytes strictly belong to amorphous state, and have made the necessary changes to our corresponding descriptions.

The following are our point-by-point responses to the questions/comments raised by each reviewer. The revisions made in the revised manuscript and supplementary information are indicated and highlighted in yellow for clarity and ease of reference.

We deeply appreciate the opportunity of revising the manuscript.

Many thanks again!

Point-by-Point Response to Reviewer #1 (R1):

General comment: The authors talk about the synthesis, characterization, and test of new glassy electrolytes of $x\text{Li}_2\text{O-TaCl}_5$ and $x\text{Li}_2\text{O-HfCl}_4$. The structure is well characterized, the Li-ion conductivity is superior, and the battery performance is attractive. I thus recommend the manuscript to Nature Comm. upon addressing the following questions:

Response: We really thank the reviewer for recognizing the importance of this work and positive recommendation of publishing. We believe that the quality of our manuscript has been significantly improved after our revisions based on these constructive suggestions.

1. The operational stack pressure should be mentioned for all batteries. Only the fabrication pressures are mentioned in Method for now.

Response: We highly thank the reviewer for this suggestion. The operational stack pressure used for all the batteries in this manuscript was ~80 MPa. This value is within the range of a typical stack pressure for inorganic electrolyte-based all-solid-state batteries (*Science*, **2021**, 373, 1494-1499 (using 50 MPa); *Nat. Energy* **2022**, 7, 83-93 (using ~250 MPa)).

Revision made: We have added the stack pressure information in the Method part of the revised manuscript.

Page 22 in the revised manuscript:

‘...The obtained internal pellet cell was sandwiched between two stainless-steel rods as current collectors. Finally, a stack pressure of ~80 MPa was applied to the solid cell for various electrochemical tests. All cell fabrication processes were carried out in an Ar-filled glove box (H_2O , $\text{O}_2 < 0.1$ ppm)...’

2. It's mentioned LCO is used in Method, but all batteries I see are by NMC83.

Response: We thank the reviewer for this comment. Two types of cathode materials were used in this work: NMC83 and LCO. NMC83 ASSBs were demonstrated in **Fig. 5a-e** and **Supplementary Fig. 17**, as well as NMC83 pouch cell in **Supplementary Fig. 19**. The performance of LCO ASSBs were exhibited in **Fig. 5f-g** and **Supplementary Fig. 18**. We double-

checked the whole manuscript and supplementary information files. Indeed, the depicted cathodes in the manuscript and supplementary information were not very visually distinct. In order to highlight the different cathode materials used in each battery, we added detailed cell configurations in Fig. 5 in the revised manuscript. We also bolded the text of “LCO” in Supplementary Fig. 18 of the revised supplementary information.

Revision made: We added detailed cell configurations in Fig. 5 in the revised manuscript. We also bolded the text of “LCO” in Supplementary Fig. 18 of the revised supplementary information.

Page 18 in the revised manuscript:

Fig. 5 | Electrochemical performances of ASSBs using $1.6\text{Li}_2\text{O-TaCl}_5$ or $1.5\text{Li}_2\text{O-HfCl}_4$ amorphous SEs.

Page 11 in the revised supplementary information:

Supplementary Fig. 18 Rate performance of LCO ASSBs using (a) $1.6\text{Li}_2\text{O-TaCl}_5$ and (b) $1.5\text{Li}_2\text{O-HfCl}_4$ SEs at -10°C .

3. More details should be provided for NMC83, purchased? SC v.s. PC.

Response: We thank the reviewer for this question. NCM83 ($\text{LiNi}_{0.83}\text{Co}_{0.11}\text{Mn}_{0.06}\text{O}_2$) cathode active material we used in this work is provided by China Automotive Battery Research Institute Co, Ltd. They are polycrystalline (PC) particles with a size around $3\ \mu\text{m}$. The SEM images of the NCM83 particles are shown in **Figure R1**.

Figure R1 SEM images of the NCM83 cathode material.

Revision made: We have added detailed information of NCM83 in the Method part of the revised manuscript.

Page 22 in the revised manuscript:

‘...10 mg of amorphous SE/NCM83 composite (3:7 mass ratio) was uniformly spread onto the surface of the one side of electrolyte layer and pressed with ~360 MPa for 5 minutes. NCM83 (LiNi_{0.83}Co_{0.11}Mn_{0.06}O₂) cathode material (polycrystalline particle size: ~3 μm) was provided by China Automotive Battery Research Institute Co, Ltd. Subsequently, Li-In alloy was placed on the other side of the...’

4. It's nice to see a pouch cell in SI. But more details should be provided, as it's not mentioned in Method, i.e., cathode loading, separator thickness, slurry casting?, layer by layer?, operational pressure, sealing, etc

Response: We highly thank the reviewer for this comment. During the revision process, we have added the detailed information about the pouch cell fabrication in the Method part of the revised manuscript.

Revision made: We have added detailed information about the pouch cell fabrication and one reference in the revised manuscript.

Page 22-23 in the revised manuscript:

‘Fabrication of all-solid-state pouch cell

The all-solid-state pouch cell was fabricated by stacking layers of NCM83/1.6Li₂O-TaCl₅ cathode, SE separators (Li₃YCl₆ and 1.6Li₂O-TaCl₅), and Li-In alloy anode. The membranes of cathode composites and SEs were made by dry-film processing method⁵⁹, where 0.5 wt% Polytetrafluoroethylene (PTFE) were added to induce the formation of doughs and followed by calendaring to the target thickness (~80 μm). The loading of the cathode was 13.125 mg cm⁻². Stacking each layers was completed in the Ar-filled glovebox, which was then sealed in plastic vacuum bag and for transferring to a dry room for further packing in the aluminum-plastic bag. A pressure of ~10 MPa was applied on the pouch cell during the cycling performance test using Neware battery testing system.’

Page 30 in the revised manuscript:

Reference:

59 Wang, C. H. *et al.* Solvent-free approach for interweaving freestanding and ultrathin inorganic solid electrolyte membranes. *ACS Energy Lett.* **7**, 410-416 (2022).

5. It's interesting to see that E_{a_LT} is smaller than E_{a_HT} , and the authors mentioned structure disorder as the contributor. Is the effect static or dynamic (e.g. "Advanced Materials, 2207411 (2022))"? A brief discussion of the mechanism along this direction will further benefit the readers.

Response: We highly appreciate the reviewer for the question and suggestion!

^7Li spin-lattice relaxation (SLR) NMR measurements can provide the activation energies not only corresponding to short-range (local motions) but also the long-range lithium-ion diffusions in bulk electrolytes. In this method, the SLR rate ($R = 1/T_1$, T_1 is the spin-lattice relaxation time) is recorded as a function of the temperature which depends on the fluctuation of the local magnetic field resulting from the motions of Li ions. These fluctuations are described by correlation function $G(t)$ characterizing the correlation time, which is related to the Li-ion jump rates between two successive jumps. Most importantly, the SLR rate is related to the $G(t)$ through the spectral density function which is the Fourier transform of $G(t)$ (*J. Electroceramics* **2017**, *38*, 142–156; *MRS Bull.* **2009**, *34*, 915–922).

In general, the diffusion-induced rate increases with increasing temperature (low-temperature flank) until it reaches a maximum followed by a decrease of R with further increasing temperature (high-temperature flank) (**Fig. 2f** and **g** in the manuscript). Since the correlation time exhibits an Arrhenius behavior, the slopes of the plot of $1/T_1$ vs $1/T$ can be used to determine the activation energies for Li-ion transport in long-range (E_{a}^{HT}) and short-range (E_{a}^{LT}) from the high-temperature flank and low-temperature flank, respectively (*ChemPhysChem* **2012**, *13*, 53–65). The E_{a}^{LT} characterizes elementary jump processes, while E_{a}^{HT} probes long-range or “bulk” Li-ion diffusions. (*J. Am. Chem. Soc.* **2016**, *138*, 11192–11201; *Phys. Chem. Chem. Phys.* **2013**, *15*, 7123–7132)

The Bloembergen, Purcell, and Pound (BPP) model used to study the fluctuation of the internal fields predicts a symmetric rate peak with an asymmetry parameter, $\beta = 2$ which does not consider the correlation effects such as Coulomb interactions (repulsive/attractive) and/or structural disorders. The E_{a}^{HT} and E_{a}^{LT} are related to each other by a relationship $E_{a}^{\text{LT}} = (\beta - 1) E_{a}^{\text{HT}}$ where $1 < \beta \leq 2$ (*Phys. Rev.* **1948**, *73*, 679–712). In the present cases, β values were

determined to be 1.34 and 1.68 for the 1.6Li₂O-TaCl₅ and the 1.5Li₂O-HfCl₄ samples, respectively, showing that the E_a^{LT} is lower than E_a^{HT} for each case. Asymmetric rate peaks are often found for structurally complex ion conductors (*Phys. Chem. Chem. Phys.* **2019**, *21*, 8489–8507; *J. Phys. Chem. Lett.* **2013**, *4*, 2118–2123; *ACS Appl. Mater. Interfaces* **2018**, *10*, 33296–33306). The deviation from the BPP model often occurs in the low-temperature regime as the local hopping of Li ions is affected by the correlation effects, leading to a reduced E_a^{LT} value comparatively (*ChemPhysChem* **2012**, *13*, 53–65).

Deviation from the BPP model ($1 < \beta < 2$) can be explained by correlation effects such as Coulomb interactions (repulsive/attractive), correlated ion dynamics, and/or structural disorders. For the amorphous oxychloride samples 1.6Li₂O-TaCl₅ and 1.5Li₂O-HfCl₄, the Li ions are exposed to an irregularly formed time-dependent potential landscape while diffusing. However, due to the challenges of completely understanding the structure of amorphous solids, we currently may not be able to elaborate on the correlation effects towards the β value deviation. Therefore, broadly speaking, we ascribed the reason for the lower E_a^{LT} to the inherited structural disorder.

The paper mentioned by the reviewer (*Adv. Mater.* **2022**, *34*, 2207411) reported that in a typical sulfide electrolyte, Li-ion conduction can be boosted by the anharmonic coupling of low-frequency Li phonon modes with high-frequency anion stretching or flexing phonon modes. The results were obtained by ab initio computation in a temperature range of 0-300 K. Comparing to the paddle-wheel phenomenon that mostly analyzed under high temperature, the results talked about in this paper (*Adv. Mater.* **2022**, *34*, 2207411) is more relevant to the practical operating temperatures of solid-state batteries. However, in our studies about ⁷Li SLR-NMR for SEs, any couplings of the spins with phonons or conduction electrons were considered negligible (*J. Am. Chem. Soc.* **2022**, *144*, 1795–1812; *Phys. Rev. B* **2008**, *77*, 024311). Therefore, from this perspective, it is hard to directly link our ⁷Li SLR-NMR results with phonon coupling to explain the factors towards the Li-ion transport, but we still can provide some additional discussion to clarify our adopted method and results.

Revision made: We have updated the following sentences and references in the revised manuscript.

Page 8-9 in the revised manuscript:

‘...In the present cases, β values were determined to be 1.34 and 1.68 for the 1.6Li₂O-TaCl₅ and the 1.5Li₂O-HfCl₄ samples, respectively, indicating structurally complex Li-ion conduction.⁴⁰⁻⁴² Generally, correlation effects (e.g., Coulomb interactions, correlated ion dynamics, structural disorders, etc.) are considered closely associated with the impacted Li-ion conduction.^{43,44} In our

studies of ^7Li SLR NMR for the amorphous SEs, the native structural disorder of the two amorphous samples was regarded as the major contributor towards the deviation of β value off 2 ($1 < \beta < 2$), leading to smaller E_a^{LT} values compared to those of E_a^{HT} .^{33,42} Elaboration on the relevant correlation effects of locally disordered structure on the Li-ion migration is proposed as an interesting research direction that appeals to further attention.'

Page 28-29 in the revised manuscript:

Reference:

- 43 Kuhn, A. *et al.* Li self-diffusion in garnet-type $\text{Li}_7\text{La}_3\text{Zr}_2\text{O}_{12}$ as probed directly by diffusion-induced ^7Li spin-lattice relaxation NMR spectroscopy. *Phys. Rev. B* **83**, 094302 (2011).
- 44 Xu, Z. M., Chen, X., Zhu, H. & Li, X. Anharmonic Cation-Anion Coupling Dynamics Assisted Lithium-Ion Diffusion in Sulfide Solid Electrolytes. *Adv. Mater.* **34**, 2207411 (2022).

Point-by-Point Response to Reviewer #2 (R2):

1: The composition of the present study is, for example, $\text{Li}_{3.2}\text{TaCl}_5\text{O}_{1.6}$. Thus, this material can be regarded as a series of oxide-ion-doped chloride-based solid electrolytes.

In such a sense, how about the conductivity of the material in the LiCl-TaCl_5 system? By comparing the behavior of the LiCl-TaCl_5 system, the effect of oxygen incorporation can be discussed.

Response: We highly appreciate the reviewer for the question, which deepens the mechanism understanding of this manuscript. During the revision, we synthesized a series of Li-Ta-Cl materials (LiTaCl_6 , Li_2TaCl_7 , $\text{Li}_{3.2}\text{TaCl}_{8.2}$, and Li_4TaCl_9) following the same experimental procedure for $x\text{Li}_2\text{O-TaCl}_5$. The stoichiometric amounts of precursors TaCl_5 and LiCl were milled using a high-speed ball milling machine at 500 rpm for 10 h. **Figure R2** shows the XRD patterns of the as-prepared Li-Ta-Cl system. Except for the LiCl and TaCl_5 impurities, the as-synthesized Li-Ta-Cl phase seemed to be amorphous. With an increasing LiCl content, the diffraction peak of the LiCl impurity became more prominent. In comparison, the Li-Ta-O-Cl system can remain amorphous for a similar range of Li/Ta ratios of 2.2 to 3.6 for $x\text{Li}_2\text{O-TaCl}_5$ ($x = 1.1-1.8$). We further obtained the ionic conductivities of above compounds by electrochemical impedance spectroscopy (EIS) method (**Figure R3**). The room-temperature (RT) ionic conductivities of LiTaCl_6 , Li_2TaCl_7 , $\text{Li}_{3.2}\text{TaCl}_{8.2}$, and Li_4TaCl_9 were 8.81×10^{-8} , 2.81×10^{-7} , 3.92×10^{-7} , and $7.20 \times 10^{-7} \text{ S cm}^{-1}$, respectively. Apparently, increasing the molar ratio of LiCl in the starting materials helped to improve the overall ionic conductivity of the Li-Ta-Cl samples. However, the values were still limited to the $10^{-7} \text{ S cm}^{-1}$ level, which were 4 orders of magnitude lower than those of the Li-Ta-O-Cl system with O incorporation. Furthermore, the activation energies of Li-Ta-Cl materials were all above 0.5 eV (**Figure R4**), which were much higher than those of the Li-Ta-O-Cl amorphous SEs.

Figure R2 Lab-based XRD patterns for the as-prepared LiTaCl_6 , Li_2TaCl_7 , $\text{Li}_{3.2}\text{TaCl}_{8.2}$, and Li_4TaCl_9 samples. The hump between 10° and 30° is due to the Kapton film which was used to protect the samples from air exposure.

Figure R3 Nyquist plots for the LiTaCl_6 , Li_2TaCl_7 , $\text{Li}_{3.2}\text{TaCl}_{8.2}$, and Li_4TaCl_9 pellets at various temperatures.

Figure R4 Arrhenius plots of the LiTaCl_6 , Li_2TaCl_7 , $\text{Li}_{3.2}\text{TaCl}_{8.2}$, and Li_4TaCl_9 samples.

Based on above results, we generally conclude three main effects of incorporating O in our Li-Ta-O-Cl amorphous SEs:

1) Moderate amount of O incorporation benefits to rearrange the Li-Ta-O-Cl frameworks and contributes to the amorphization of Li-Ta-O-Cl. Similar result can also be found in the Si_2O -doped $\text{Li}_2\text{S}-\text{B}_2\text{S}_3-\text{LiI}$ system (*Adv. Energy Mater.* **2020**, *10*, 1902783).

2) Partially replacing Cl with O can dramatically increase the ionic conductivities in Li-Ta-O-Cl system. This is probably related to the strong electronegativity of O anions. We already proved in our manuscript that O mainly acts as bridges to connect Ta-centered polyhedrons (Fig. 3 in the manuscript). The electron distribution of O-Ta-Cl in Li-Ta-O-Cl materials is more asymmetric than the Cl-Ta-Cl in Li-Ta-Cl materials, leading to weak Coulombic forces between terminal Cl and Li. As a result, Li ions could be easier to escape from one site and jump to another. Besides, oxygens at the bridging positions can enlarge the doorway radius for easy access of Li ions (*J. Phys. Chem. B* **2006**, *110*, 16318-16325).

3) O substitution decreases the activation energies of Li-Ta-Cl. This is because the bridging oxygen networks can induce a much wider range of distortions in Li sites, which are the predominance to realize a Li-ion energy landscape with low migration energy (*Nat. Mater.* **2022**, *21*, 1-8).

In the current manuscript, we explained the benefits of oxygen incorporation (see below the quote/paste sentences from the last paragraph of Page 14 in the manuscript).

‘...corner-shared oxygen (O-2Ta) networks could induce a much wider range of distortions in Li sites.⁵⁷ The distorted lithium sites in O-2Ta networks were the predominance to realize a Li-ion energy landscape with low migration energy. At the same time, oxygens at the bridging positions would enlarge the doorway radius for easy access of Li ions.⁵⁸ Third, the unsaturated Ta–Cl···Li bonds in $[\text{TaCl}_{5-a}\text{O}_a]^{3-}$ ($1 \leq a < 5$) showed weak Coulombic forces between lithium and chlorine, making it easier for Li ions escape from one site and jump to another...’

In order to emphasize and fully support this point, we updated the descriptions in the revised manuscript and added the XRD and EIS data of chloride-based $\text{Li}_{3.2}\text{TaCl}_{8.2}$ in the revised supplementary information.

Revision made: We have added one figure in the revised supplementary information as **Supplementary Fig. 12**. The following sentences have been updated in the revised manuscript.

Page 14 in the revised manuscript:

‘...making it easier for Li ions escape from one site and jump to another. **In short, oxygen incorporation is beneficial to the amorphization of $x\text{Li}_2\text{O-TaCl}_5$. The induced disordered structures in amorphous $x\text{Li}_2\text{O-TaCl}_5$ lead to a sharply increased ionic conductivity and decreased activation energy compared to the single-anion Li-Ta-Cl sample (see comparison data in **Supplementary Fig. 12**).**’

Page 8-9 in the revised supplementary information:

Supplementary Fig. 12 A Li-Ta-Cl sample without O incorporation was prepared following the same experimental procedure for $x\text{Li}_2\text{O-TaCl}_5$. The composition of $\text{Li}_{3.2}\text{TaCl}_{8.2}$ was chosen for the same Li/Ta ratio as the most conductive $1.6\text{Li}_2\text{O-TaCl}_5$ amorphous SE: (a) Lab-based XRD patterns for the as-prepared $\text{Li}_{3.2}\text{TaCl}_{8.2}$. The hump between 10° and 30° is the diffraction peak of the Kapton film which was used to protect the sample from air exposure. The diffraction peaks can be assigned to the TaCl₅ and LiCl raw materials. (b) Nyquist plots and (c) Arrhenius plot for

the $\text{Li}_{3.2}\text{TaCl}_{8.2}$ pellet at various temperatures. Overall, without O incorporation, the completed amorphization of Li-Ta-Cl would be difficult. The RT ionic conductivity of $\text{Li}_{3.2}\text{TaCl}_{8.2}$ dropped to the order of $10^{-7} \text{ S cm}^{-1}$ with a significantly increased activation energy of 0.546 eV.

2. How the authors determined that the observed conductivity is “ionic conductivity”? The author should show the transport number of electronic conduction, ionic conduction, and Li-ion conduction.

Response: We sincerely appreciate the reviewer for this question, which makes the data description in the revised manuscript more accurate. Indeed, the value derived from the EIS test could be roughly regarded as ionic conductivity only if the charge carriers in the tested SEs are Li ions (*Acc. Mater. Res.* **2021**, 2, 869–880). In our manuscript, other characterizations are needed to prove the reported amorphous SEs are not only electron insulators but also good Li-ion conductors. During the revision, we performed direct current (DC) measurements of $1.6\text{Li}_2\text{O-TaCl}_5$ and $1.5\text{Li}_2\text{O-HfCl}_4$ pellets under ion-blocking condition (**Figure R5a**) and electron-blocking condition (**Figure R5f**) to calculate the effective electronic and Li-ion conductivities, respectively (*Energy Environ. Sci.*, **2023**, 16, 610; *Adv. Mater.* **2018**, 30, 1803075). As shown in **Figure R5b** and **d**, a constant voltage was applied to an ion-blocking symmetric cell for an hour until the cell polarization reached equilibrium. The stabilized current responses were then recorded at different voltages from 0.1 V to 0.5 V (**Figure R5c** and **e**). The electron conductivity (σ_e) of $1.6\text{Li}_2\text{O-TaCl}_5$ or $1.5\text{Li}_2\text{O-HfCl}_4$ can be determined by **Equation 1**:

$$\sigma_k = \frac{d}{AR_k} \quad (k = e^- \text{ or } \text{Li}^+) \quad \text{Equation 1}$$

where d is the thickness of the $1.6\text{Li}_2\text{O-TaCl}_5$ or $1.5\text{Li}_2\text{O-HfCl}_4$ pellet (between 0.04 cm and 0.06 cm), A is the geometric area of the pellet, and R_e can be obtained via Ohm’s Law from **Figure R5c** and **e**. As a result, the electronic conductivity was $3.37 \times 10^{-10} \text{ S cm}^{-1}$ for $1.6\text{Li}_2\text{O-TaCl}_5$ and $1.57 \times 10^{-10} \text{ S cm}^{-1}$ for $1.5\text{Li}_2\text{O-HfCl}_4$, which were seven orders of magnitude lower than the EIS conductivity values. Therefore, electron transport in both $1.6\text{Li}_2\text{O-TaCl}_5$ and $1.5\text{Li}_2\text{O-HfCl}_4$ can be considered negligible.

We further verified the conducting carriers in $1.6\text{Li}_2\text{O-TaCl}_5$ and $1.5\text{Li}_2\text{O-HfCl}_4$ via the similar DC measurement but with a different cell configuration (**Figure R5f**). In this case, $1.6\text{Li}_2\text{O-TaCl}_5$ or $1.5\text{Li}_2\text{O-HfCl}_4$ was sandwiched by $\text{Li}_6\text{PS}_5\text{Cl}$ SE and Li metal. Since $\text{Li}_6\text{PS}_5\text{Cl}$ is known as a Li-ion conductor, this cell configuration can block electrons as well as other ion

species (such as Cl and O ions) and only allow the access of Li ions. As shown in **Figure R5g-j**, the polarized current response was recorded under different bias voltages. The total Li-ion resistance (R_{Li^+}) can be obtained as 65.12 Ω for 1.6Li₂O-TaCl₅-contained cell and 99.09 Ω for 1.5Li₂O-HfCl₄-contained cell. To subtract the additional resistance contributions from the Li₆PS₅Cl SE and Li metal, a symmetric cell of Li/Li₆PS₅Cl/Li was also constructed ($R_{Li_6PS_5Cl\ SE + Li} = 53.87\ \Omega$). Finally, a DC Li-ion conductivity (σ_{Li^+}) was calculated to be $6.35 \times 10^{-3}\ S\ cm^{-1}$ for 1.6Li₂O-TaCl₅ and $1.67 \times 10^{-3}\ S\ cm^{-1}$ for 1.5Li₂O-HfCl₄, which was consistent with those from the EIS measurement. The small differences (1.6Li₂O-TaCl₅: $6.35 \times 10^{-3}\ S\ cm^{-1}$ (DC) vs. $6.6 \times 10^{-3}\ S\ cm^{-1}$ (AC); 1.5Li₂O-HfCl₄: $1.67 \times 10^{-3}\ S\ cm^{-1}$ (DC) vs. $1.97 \times 10^{-3}\ S\ cm^{-1}$ (AC)) may be caused by different cell configurations or the interfacial resistance contribution between the xLi₂O-MCl_y and the Li₆PS₅Cl layers.

Based on above results, we can identify that xLi₂O-MCl_y (M = Ta or Hf, $0.8 \leq x \leq 2$, y = 5 or 4) amorphous SEs are pure Li-ion conductors. The observed ionic conductivity via EIS test can be regarded as the Li-ion conductivity for each of the compound.

Figure R5 Electronic conductivity (a-e) and ionic conductivity (f-j) determination by DC measurements at room temperature. (a) Cell configuration, (b, d) DC polarization curves, and (c, e) equilibrium current response for $1.6\text{Li}_2\text{O-TaCl}_5$ and $1.5\text{Li}_2\text{O-HfCl}_4$ symmetric cell under ion-blocking condition. Similarly, (f) shows the electron-blocking cell configuration while (g-j) present the DC polarization results.

Revision made: In response to the reviewer's concern, we have added the **Figure R5** and relevant description in the revised supplementary information as **Supplementary Fig. 7**. The following descriptions, measurements, and references have also been updated in the revised manuscript.

Page 7 in the revised manuscript:

‘...The $1.5\text{Li}_2\text{O-HfCl}_4$ SE in mostly amorphous state showed the highest ionic conductivity ($1.97 \times 10^{-3} \text{ S cm}^{-1}$) and a low activation energy (0.328 eV) among the $x\text{Li}_2\text{O-HfCl}_4$ series (**Fig. 2d** and **Supplementary Fig. 6**). Direct current (DC) measurements for the representative $1.6\text{Li}_2\text{O-TaCl}_5$ and $1.5\text{Li}_2\text{O-HfCl}_4$ amorphous SEs under ion-blocking and electron-blocking conditions^{13,32} were also conducted as shown in **Supplementary Fig. 7**. The determined electronic conductivities were negligible (at $10^{-10} \text{ S cm}^{-1}$ order). The Li-ion conductivities calculated from the DC measurements agree well with the values we derived from the EIS measurements, confirming the $x\text{Li}_2\text{O-MCl}_y$ amorphous SEs as excellent Li-ion conductors.’

Page 27 in the revised manuscript:

Reference:

13 Asano, T. et al. Solid halide electrolytes with high lithium-ion conductivity for application in 4 V class bulk-type all-solid-state batteries. *Adv. Mater.* **30**, 1803075 (2018).

32 Kwok, C. Y., Xu, S. Q., Kochetkov, I., Zhou, L. D. & Nazar, L. F. High-performance all-solid-state Li₂S batteries using an interfacial redox mediator. *Energy Environ. Sci.* **16**, 610-618 (2023).

Page 21 in the revised manuscript:

‘Measurements of ionic and electronic conductivities:

...The applied frequency range was 1 Hz ~ 7 MHz and the voltage amplitude was 20 mV. The cell assembly process for DC measurements was similar with that for EIS test. To determine the electronic conductivity, the current responses of the cell was measured at a range of constant voltages for 60 min each. The applied voltage ranged from 0.1 to 0.5 V with a step size of 0.1 V. The DC Li-ion conductivity was evaluated with a symmetric cell configuration of Li/Li₆PS₅Cl/ $x\text{Li}_2\text{O-MCl}_y$ /Li₆PS₅Cl/Li under a bias voltage for 30 min. The bias voltage was applied at 5, 10, 15, 20, and 25 mV consecutively. The Li₆PS₅Cl SE (provided by China Automotive Battery Research Institute Co, Ltd) was used to prevent direct contact between Li metal and $x\text{Li}_2\text{O-MCl}_y$ SE.’

Supplementary Fig. 7 Electronic conductivity (a-e) and ionic conductivity (f-j) determination by DC measurements at room temperature. (a) Cell configuration, (b, d) DC polarization curves, and (c, e) equilibrium current response for 1.6Li₂O-TaCl₅ and 1.5Li₂O-HfCl₄ symmetric cell under ion blocking condition. (f) The electron-blocking cell configuration and (g-j) the DC polarization results for Li-ion conductivity evaluations of 1.6Li₂O-TaCl₅ and 1.5Li₂O-HfCl₄. Li₆PS₅Cl was

chosen as an interlayer SE since it is a recognized Li-ion conductive SE and kinetically stable with Li metal. The electronic conductivities of $1.6\text{Li}_2\text{O-TaCl}_5$ and $1.5\text{Li}_2\text{O-HfCl}_5$ were $3.37 \times 10^{-10} \text{ S cm}^{-1}$ and $1.57 \times 10^{-10} \text{ S cm}^{-1}$, respectively. The DC measurements derived that Li-ion conductivities of $6\text{Li}_2\text{O-TaCl}_5$ and $1.5\text{Li}_2\text{O-HfCl}_5$ were $6.35 \times 10^{-3} \text{ S cm}^{-1}$ and $1.67 \times 10^{-3} \text{ S cm}^{-1}$, respectively.

3. In the Figures for charge-discharge curves, the cell configuration should be shown, e.g. *Li-In/Li3YCl6/xLi2O-TaCl5/LiCoO2*.

Response: We highly appreciate the reviewer for this comment. During the revision process, we have added the cell configurations in the corresponding figures of the revised manuscript.

Revision made: We have added all the cell configuration information in **Fig. 5** of the revised manuscript.

Page 18 in the revised manuscript:

Fig. 5 | Electrochemical performances of ASSBs using $1.6\text{Li}_2\text{O-TaCl}_5$ or $1.5\text{Li}_2\text{O-HfCl}_4$ amorphous SEs.

4. In the low-temperature charge-discharge test, the authors described that LGPS was used. The cell configuration was Li-In/LGPS/xLi₂O-TaCl₅/LiCoO₂?

Anyway, the electrochemical window of the present solid electrolytes should be discussed.

Response: We highly appreciate the reviewer for this comment. In the low-temperature charge-discharge test, we used LGPS interlayer between 1.6Li₂O-TaCl₅ and Li-In anode to prevent the direct contact of 1.6Li₂O-TaCl₅ and Li-In anode, and at the same time, decrease the total cell resistance at the low-temperature testing environment. We have added the cell configuration information (such as Li-In/Li₃YCl₆/xLi₂O-MCl_y/NCM83 and Li-In/LGPS/xLi₂O-MCl_y/LCO) in the corresponding figures (see the response for Question 3 from Reviewer #2).

We totally agree with the reviewer that electrochemical stability window (ESW) of an SE is important and should be provided. During the revision process, we tested the ESWs of two representative amorphous SEs: 1.6Li₂O-TaCl₅ and 1.5Li₂O-HfCl₄. As shown in **Figure R6**, the ESWs of 1.6Li₂O-TaCl₅ and 1.5Li₂O-HfCl₄ are 2.20 V–4.15 V and 2.10 V–4.10 V (vs. Li⁺/Li). Since the cathodic limit of both two types of amorphous SEs is higher than 2 V, we choose Li₃YCl₆ or LGPS as interlayer to separate amorphous SE and Li-In anode in ASSBs. Similar strategies have been widely used for the halide-based ASSBs (*Energy Environ. Sci.*, **2020**, *13*, 2056–2063; *Nat. Energy* **2022**, *7*, 83–93; *Nat. Commun.* **2021**, *12*, 4410).

Figure R6 Linear cyclic voltammetry (LSV) profiles of (a) 1.6Li₂O-TaCl₅ and (b) 1.5Li₂O-HfCl₄. (The working electrode consists of 80 wt % SE and 20 wt % carbon black).

Revision made: In response to the reviewer's concerns, we have added the **Figure R6** in the revised supplementary information as **Supplementary Fig. 16**. The following sentences and LSV test method have also been updated in the revised manuscript.

Page 16 in the revised manuscript:

‘...As promising SEs for bulk-type all-solid-state batteries (ASSBs), the electrochemical stability windows of $1.6\text{Li}_2\text{O-TaCl}_5$ and $1.5\text{Li}_2\text{O-HfCl}_4$ amorphous SEs were measured (**Supplementary Fig. 16**). Although the amorphous SEs were thermodynamically unstable against metal anodes (such as Li and Li-In) due to the reduction of Ta^{5+} or Hf^{4+} , they showed good oxidative stability and are promising to tolerate cathode active materials beyond 4 V. The electrochemical performances of $1.6\text{Li}_2\text{O-TaCl}_5$ and $1.5\text{Li}_2\text{O-HfCl}_4$ amorphous SEs were then evaluated with $\text{LiNi}_{0.83}\text{Co}_{0.11}\text{Mn}_{0.06}\text{O}_2$ (NCM83) and LiCoO_2 (LCO).’

Page 21-22 in the revised manuscript:

‘Linear sweep voltammetry (LSV) test:

Approximately 80 mg of the $1.6\text{Li}_2\text{O-TaCl}_5$ or $1.5\text{Li}_2\text{O-HfCl}_4$ amorphous SE powder was cold-pressed into a pellet. A 10-mg mixture of amorphous SE and carbon black (CB) (8:2 wt./wt.) was uniformly covered on one side of the pellet as working electrode. Li foil was attached on the other side of the pellet as both counter and reference electrode. A $\text{Li}_6\text{PS}_5\text{Cl}$ interlayer (~40 mg) was adopted to avoid the incompatibility between amorphous SE and metallic Li. The LSV measurements were conducted using a versatile multichannel potentiostat 3/Z (VMP3) with a positive scan range from open-circuit voltage (OCV) to 6 V and a negative scan range from OCV to 0 V. The scan rate was 0.1 mV s^{-1} .’

Page 10 in the revised supplementary information:

Supplementary Fig. 16 Linear cyclic voltammetry (LSV) profiles of (a) $1.6\text{Li}_2\text{O-TaCl}_5$ and (b) $1.5\text{Li}_2\text{O-HfCl}_4$.

5. The authors use the expression “glassy”. However, the authors only show that the materials are amorphous by XRD pattern and no evidence for a “glassy state.” In the strict sense, “glass” and “amorphous” is not the same category of material, and “glassy” material should show the glass transition behavior. The reviewer strongly recommends using “amorphous” in the present study

Response: We highly appreciate the reviewer for this comment, which makes the expression in the current manuscript more accurate. We detailly checked the literatures during the revision process. Indeed, “glass” refers to a solid which not only is non-crystalline, but also shows glass transition behavior (*Solid State Ionics* **1990**, 38, 217-224; *Solid State Ionics* **1996**, 86-88, 487-490). The commonly reported glassy SEs include Li₂S-P₂S₅, Li₂S-SiS₂, Li₂O-B₂O₃, etc (*Electrochim. Acta* **2006**, 52, 1576–1581; *Adv. Mater.* **2005**, 17, 918-921). While for an amorphous solid, it is a broader concept, which indicates that the solid lacks long-range ordered structures (*J. Am. Ceram. Soc.* **2001**, 84, 477–479). Considering the non-crystalline nature of our xLi₂O-MCl_y SEs that we have revealed via XRD measurements, we fully agree with the reviewer that “amorphous” is a more accurate and proper word to describe the SEs in this manuscript.

Revision made: We have updated the entire manuscript by replacing “glassy/glasses” with “amorphous/amorphous SEs”. The changes have been highlighted.

Again, many thanks for your constructive suggestions and comments!

REVIEWERS' COMMENTS

Reviewer #2 (Remarks to the Author):

The reviewer believes that the revised manuscript can be accepted for publication.